# Flow Equations for Free-Flowable Particle Fractions of Sorbitol for Direct Compression: An Exploratory Multiple Regression Analysis of Particle and Orifice Size Influence

**DOI:** 10.3390/pharmaceutics14081653

**Published:** 2022-08-08

**Authors:** Julia Marushka, Hana Hurychová, Zdenka Šklubalová, Jurjen Duintjer Tebbens

**Affiliations:** 1Faculty of Pharmacy in Hradec Kralove, Department of Pharmaceutical Technology, Charles University, 500 05 Hradec Kralove, Czech Republic; 2Faculty of Pharmacy in Hradec Kralove, Department of Biophysics and Physical Chemistry, Charles University, 500 05 Hradec Kralove, Czech Republic

**Keywords:** mass flow rate, flow equation, powders properties, orifice diameter, hopper, particle size, multilinear regression, interaction term

## Abstract

**Highlights:**

**Abstract:**

Flowability is among the most important properties of powders, especially when fine particle size fractions need to be processed. In this study, our goal was to find a possibly simple but accurate mathematical model for predicting the mass flow rate for different fractions of the pharmaceutical excipient sorbitol for direct compression. Various regression models derived from the Jones–Pilpel equation for the prediction of the mass flow rate were investigated. Using validation with experimental data for various particle and hopper orifice sizes, we focused on the prediction accuracy of the respective models, i.e., on the relative difference between measured and model-predicted values. Classical indicators of regression quality from statistics were addressed as well, but we consider high prediction accuracy to be particularly important for industrial processing in practice. For individual particle size fractions, the best results (an average prediction accuracy of 3.8%) were obtained using simple regression on orifice size. However, for higher accuracy (3.1%) in a unifying model, valid in the broad particle size range 0.100–0.346 mm, a fully quadratic model, incorporating interaction between particle and orifice size, appears to be most appropriate.

## 1. Introduction

Most pharmaceutical powders are fine to coarse powders varying in shape and physical properties due to the specific production technologies (crystalized, milled, agglomerated, granulated, etc.). This influences their flow, consolidation, compression, and compaction properties. As the flowability of powder excipients and granules is an important parameter for transport, handling, and storage in the pharmaceutical industry [1,2], accurate characterization of the flow properties is necessary [3,4].

For evaluation of the flow rate of particulate material, one generally prefers to assess flow through the orifice of a hopper when compared with different settings [5,6], and mass flow rate is the recommended quantity to measure for free flowable bulk solids. It is abundantly used in the description of the discharge of powders from a hopper into the matrix of the tablet press or capsules during filling [7,8]. Cohesive forces between small particles may cause problems (ratholing, arching, etc.), and correct description and quantification of flow behaviour allow for eliminating losses caused by nonuniform filling or jamming of the equipment [9,10,11,12,13]. For a large group of pharmaceutical excipients, the significant effect of granulometric characteristics on the flow behaviour (including avalanche and shear behaviour) was proved previously; see, e.g., [14,15].

It is generally accepted that the flow rate of a powder material through a hopper orifice under gravity is influenced by the orifice diameter, powder density, particle size, and shape. Flow equations can describe the relationship between these variables (see, e.g., [16,17,18,19,20,21]), with most of them being derived from the basic Formula (1) introduced by Brown and Richards [16]:(1)$$Q_{m}=\frac{\pi }{4}\;\cdot d\cdot g^{1/2}{\left(D-kX\right)}^{5/2},$$
where *Q_m_* is the mass flow rate, *d* is the powder density, *g* is the gravitational acceleration, *D* is the diameter of the hopper aperture, *k* is the empirical shape coefficient, and *X* is the particle diameter. The particle size and the diameter of the orifice are believed to be the most important factors, while the effect of particle size is often assumed to be smaller [22,23,24]. It was experimentally observed that the particles leaving the hopper show a tendency to adhere to the edge of the orifice, forming a stagnant zone along its perimeter that leads to a reduction of *D* to the effective orifice flow diameter *D* − *kX* [25]. The term *kX* is referred to as the empty annulus. The larger the particle size, the stronger this empty annular zone effect is. The shape coefficient *k* is usually in the range of 1 ≤ *k* ≤ 2 [26].

For a given orifice diameter, the influence of *X* on *Q_m_* can be described by a flow curve, in which monodisperse particles reach a maximum flow rate, followed by a decline when the particle size further increases [27,28,29,30]. Particles smaller than 0.200 mm in diameter can produce intermittent flow with high flow rate variability for several reasons: due to cohesive forces, because the air pressure gradient decreases the gravitational flow of small particles significantly for fine powders with lower air permeability [31,32] or due to an extended particle surface responsible for friction or electrostatic charge.

A flow rate equation in which the particle diameter is not a component is the Jones–Pilpel equation [33,34,35], which applies to both cylindrical and conical hoppers:(2)$$Q_{m}=\frac{\pi }{4}\;\cdot d_{b}\cdot g^{1/2}\cdot {\left(\frac{D}{a}\right)}^{n}\;,$$
where *a* and *n* are dimensionless equation parameters, and *d_b_* is the bulk density (which is here assumed more appropriate than *d*). *n* is, in general, close to the previous exponent 5/2, and *a* is used to modify *D* to the effective diameter *D*/*a*, similarly to the modification *D* − *kX* above. Clearly, in this equation, the influence of particle diameter may be implicitly present, for instance, when the bulk density or the parameter *a* is a function of it.

This seems to be the case for the present study, where we aim to characterize the mass flow rate for four fine particle size fractions of the model, free flowable pharmaceutical excipient sorbitol, for direct compression through a conical test hopper. Considering the growing interest in mathematical modelling for quantifying technological processes in the pharmaceutical industry and their complexity [36,37,38,39,40,41], in our study, we will look for a simple mathematical model for predicting the mass flow rate with acceptable high precision of its prediction. Using comprehensible linear and quadratic regression equations, special attention will be paid to the possible mutual influence of *D* and *X*. This includes the question of whether an interaction term for *D* and *X* is appropriate and an investigation of the statistical significance of the estimated effects of the respective factors.

## 2. Materials and Methods

All measurements were carried out at a standard laboratory temperature of 21 ± 1 °C and relative air humidity of 29 ± 2% (Hygrometer 608-H1, Testo, Shenzhen, China).

### 2.1. Experimental Material

Sorbitol for direct compression (Merisorb, Merisorb 200 Pharma, Tereos Syral SAS Nesle, Mesnil-Saint-Nicaise, France) was used as a model pharmaceutical excipient.

### 2.2. Scanning Electron Microscopy (SEM)

The measurement of the particle shape of raw sorbitol (Figure 1) was performed using a scanning electron microscope (Phenom Pro, Phenom-World B. V., Rotterdam, The Netherlands) with a backscattered electron detector (BSD). The samples were carefully sprinkled onto a carbon conductive tape. The images were generated at the acceleration voltage of 5 kV and a magnification of 250×.

### 2.3. Particle Size Fractions

Four particle size fractions in the ranges of 0.080–0.125, 0.125–0.200, 0.200–0.300, and 0.300–0.400 mm were obtained using a Vibratory Sieve Shaker AS 200 basic (RETSCH, Haan, Germany) in accordance with the recommendation of the European Pharmacopoeia (Ph. Eur. 10.8, 2.9.38). After assembling the sieves, 50.0 g of powder was placed on the largest sieve. The sieves were subjected to a standardized period of agitation for 5 min, and the mass of material retained on each sieve was estimated using a laboratory balance (precision of 0.01 g). The sieving continued until the powder mass on any of the test sieves did not change by more than 5%, which usually took 15 to 20 min. The particle size fractions *X* were described by the geometrical mean of the screens, namely 0.100, 0.158, 0.245, and 0.346 mm; see Table 1.

### 2.4. Measurement of Bulk Density

The bulk density of particle size fractions was determined by Scott’s volumeter (Copley, Nottingham, UK) in accordance with the European Pharmacopoeia 10.8 (2.9.34). The bulk density *d_b_* (g/mL) was calculated from the known volume of the powder (25.00 mL) and its mass. Table 1 lists the means of ten measurements with standard deviations, SD, in brackets. The correlation coefficient between the four particle size geometrical means and bulk densities were equal to 0.97 (*p*-value 0.03).

### 2.5. Measurement of Mass Flow Rate

The mass flow rate *Q_m_* of the sorbitol particle size fractions was measured using the Automated Powder and Granulate Testing System (PTG S3, Pharmatest, Hainburg, Germany) in agreement with the Ph. Eur. 10.8 (2.9.36). A stainless-steel conical hopper with a capacity of 300.0 mL having an internal angle wall inclination of 40° was used to measure the time it took to empty 50.0 g of powder through the circular aperture with diameter *D* = 6.0 mm, 8.0 mm, 10.0 mm and 15.0 mm, respectively, see Table 2. The mass flow rate *Q_m_* (g/s) was then calculated. The discharge homogeneity was checked by registering the mass/time profile at balance.

The values of the experimentally detected mass flow rates *Q_m_* for the four levels of orifice diameter and the four particle size levels are displayed in Table 3. The values are averages of ten measurement repetitions with SD in brackets. Data are complemented with the volume flow rate *Q_v_* (mL/g), calculated as follows:(3)$$Q_{v}=\frac{Q_{m}}{d_{b}}\;,$$
with the SD being that of *Q_m_* divided by the corresponding value of *d_b_.*

### 2.6. Basic Regression Model

The Jones–Pilpel Equation (2) can be transformed using the natural logarithm into
(4)$$lnQ_{pm}=n\;ln\;D+ln\left(\frac{\pi }{4}\;\cdot d_{b}\cdot \;\sqrt{g}\right)-n\;ln\;a\;,$$
where *Q_pm_* denotes the *predicted* mass flow rate, as opposed to the measured one in Table 3.

This equation can be regarded as a simple linear regression model for *ln Q_pm_* with a single variable *ln D* and the remaining two terms representing the intercept (constant term). It can be written as *ln Q_pm_ = m*_1_
*ln D + m*_0_, where the regression output obtained from our measured samples includes the slope *m*_1_, an estimate for *n*, and the intercept *m*_0_, which estimates $ln\left(\frac{\pi }{4}\;\cdot d_{b}\cdot \;\sqrt{g}\right)-n\;ln\;a$ as a whole. Because of the strong correlation between bulk density and particle size, we assume that *d_b_* is a direct function of *X*. Hence, the influence of *X* is included in *m*_0_ implicitly.

To detect the influence of *X* and *D* explicitly, we extend the model with the explanatory variable *X*. We will consider models ranging from a basic multilinear regression model,
(5)$$ln\;Q_{pm}=\;k_{1}\cdot ln\;D+\;k_{2\;}\cdot ln\;X+k_{0}\;,\;\cdots$$
to a fully quadratic model, including the interaction between *X* and *D*,
(6)$$ln\;Q_{pm}=k_{11}\cdot {\left(lnD\right)}^{2}+k_{22}\cdot \;{\left(lnX\right)}^{2}+k_{12}\cdot ln\;X\;\cdot ln\;D+k_{1}\cdot ln\;D+k_{2\;}\cdot ln\;X+k_{0}$$

Our aim, however, is to find as simple an accurate model as possible.

To evaluate the quality of the models, we primarily investigated the precision of the prediction. The precision of the prediction Δ*Q_pm_* (%) is the size of the relative difference between the value measured and that predicted:(7)$$\Delta Q_{pm}\;\left(\%\right)=\left|\frac{Q_{m}-Q_{pm}}{Q_{m}}\right|\cdot 100\%$$

We also considered the coefficient of determination, which gives the ratio of the variability explained by the regression to the overall variability in the observed data. Furthermore, we considered the statistical significance of the computed regression coefficients, with strong significance indicating that based on the measured values, one can be strongly confident that the corresponding coefficient is generally different from zero. For all regression models, we inspected the *p*-value in the ANOVA table and the normality of residuals. As both were always satisfactory, we did not report these indicators of model quality.

In our models based on Equation (4), the influence of density was replaced by that of particle size. When evaluating models with the factor particle size, we also illustrated the effect of density by displaying the volume flow rate, calculated by Equation (3). However, because *Q_v_* is just a multiple of *Q_m_*, the precision of the volume flow rate prediction is the same, and we displayed only the precision Δ*Q_pm_*.

All numbers presented in the tables are the results of measurements or computations, with the full number of digits available in Microsoft Excel (2022); the final numbers have been rounded for clear presentation. Graphs were created in MATLAB (The MathWorks, Inc., Natick, MA, USA).

## 3. Results and Discussion

The measurement of mass flow rate through a hopper orifice belongs to standard procedures in flowability testing [3,4]. Using four particle size fractions of the model, free-flowable excipient sorbitol for direct compression, we systematically investigated the influence of the orifice diameter and the particle size on the mass flow rate. Based on our laboratory data, a series of models of the form (5) and (6) listed below were inspected. For practical usage, the main criterion for usability was high prediction precision, Δ*Q_pm_*, which is the value in percentage of the mean relative difference between the experimentally observed flow rate and the one predicted by the generated model. The lower the value of Δ*Q_pm_*, the more precise the prediction is; ideally, values of Δ*Q_pm_* under 10% were expected.

### 3.1. Linear Regression for Individual Fractions

For each particle size fraction (monodisperse particles), we start by only investigating the influence of D. For every level *X_I_*–*X_IV_*, we use simple linear regression models where in (5) the coefficient *k*_2_ is fixed to zero. These yield:(8)$$lnQ_{pm}=2.34\cdot lnD-3.50,\;\mathrm{for}\;X_{I}\;$$
(9)$$lnQ_{pm}=2.47\cdot lnD-3.50,\;\mathrm{for}\;X_{II}\;$$
(10)$$lnQ_{pm}=2.61\cdot lnD-3.72,\;\mathrm{for}\;X_{III}\;$$
(11)$$lnQ_{pm}=2.67\cdot lnD-3.86,\;\mathrm{for}\;X_{IV}\;$$

The corresponding predictions are given in Table 4 (third column).

In this table (and similarly in the tables for later models), the predicted values *Q_pm_* are obtained by taking the exponential values of $lnQ_{pm}$, with the corresponding levels of *D* substituted. Information on the statistical significance of the computed coefficients can be found in the first four columns of Table 8. Graphs of the regression lines are displayed in Figure 2.

We see that the obtained models are very accurate, with the average precision Δ*Q_pm_* over the four models equal to 3.85% and the overall average coefficient of determination *R*^2^ = 0.998. The computed regression coefficients have high statistical significance. This demonstrates that, if the practical situation allows, it can be beneficial to consider narrow particle size distributions.

The estimated slopes fluctuate around the value 5/2, in accordance with the parameter *n* in Equation (2), and the computed regression lines (Figure 2) clearly depend on the considered fraction size.

### 3.2. Linear Regression Using All Levels of X Pooled

Achieving a narrow particle size distribution (monodisperse particles), e.g., by sieving, can be time-consuming and expensive, particularly in the case of fine particles. Given rather accurate models for the individual levels of *X*, we investigated whether a unified model for all particle sizes, representing a wider size distribution, can be used. We computed a single regression line using all available flow data in Table 3:(12)$$lnQ_{pm}=2.52\cdot lnD-3.64$$

Table 5 gives information on the corresponding predictions; a graphical illustration of the model is represented in Figure 3. The two computed regression coefficients, 2.52 and 3.64, have high statistical significance Table 8.

As expected, we obtained a somewhat lower coefficient of determination, *R*^2^ = 0.9621, when pooling fractions in one experimental data set. The prediction precision is 14.26% on average, i.e., the quality has dropped compared to the average of 3.85% for models (8)–(11). Looking at the prediction precision for individual particle size fractions, the poorest prediction was observed for the *X_I_* fraction with the smallest particle diameter of 0.100 mm. At the same time, the prediction precisions were influenced by the orifice size *D*, with the best results for *D* = 6 mm.

As the study of flow behaviour has a practical impact when using a material with a wider size distribution, we included the particle size *X* in the regression models investigated in the following subsections. The hope is to achieve a more accurate unifying predicting model, with precision and values of *R*^2^ closer to those for the models (8)–(11), and with a more homogenous distribution of predicted precisions.

### 3.3. Multiple Regression without Interaction

We first consider the multilinear regression model without interaction, which, for our 16 measurements, takes the form
(13)$$lnQ_{pm}=2.52\cdot lnD+0.30\cdot lnX-3.15$$

Indicators of model quality are displayed in Table 6. A 3D plot of the predicted mass flow rate in dependence on both the diameter and particle size is presented in Figure 4.

This shows that the detected influence of *X* is very slight; its slope is about ten times smaller than the slope for *D*, which is close to that estimated in the model (12). This agrees with the generally accepted opinion that the effect of orifice size is much more pronounced than that of particle size. Both slopes, however, are statistically significant (Table 8). The average prediction precision (7.49%) has clearly improved compared with the model (12).

Though *R*^2^ is higher than in the model (12), this may be due to the fact that *R*^2^ increases artificially when another independent variable is added. Wherever more than one factor is studied, the adjusted coefficient of determination (*R_A_*^2^) gives more objective information. *R_A_*^2^ takes the value of 0.9868 (Table 6), which is higher than *R*^2^ for model (12), thus suggesting that the inclusion of factor *X* increases the portion of the variability in the model’s data. However, several precision levels are above 10%. Additionally, the prediction precisions are still somewhat irregularly distributed among the individual levels of *X*; e.g., *X_I_* exhibits high precision at *D_I_* but very poor precision at *D_IV_*.

### 3.4. Multiple Regression with Interaction

When incorporating an interaction term for *D* and *X* in the multiple regression model (5), we obtain
(14)$$lnQ_{pm}=2.97\cdot ln\;D-0.31\cdot lnX+0.27\cdot ln\;X\cdot lnD-4.15,\;$$
giving Table 7 and Figure 5.

Although the surface looks like a plane with a standard angle of view (Figure 5a), it has to be twisted because the model is not linear. It is, in fact, only very slightly twisted (Figure 5b); this results from the dominance of the slope for *D*. The slope for *D*, 2.97, may seem too far from the theoretical exponent value 5/2 for *n* in Equation (2), but note that due to the interaction term 0.27·ln X·lnD, the total slope before *ln D* is (2.97 + 0.27·ln X). With *X* in the range of 0.1 to 0.346, this gives a total slope of 2.34 to 2.68. An overview of the estimates of the slopes (and intercepts) for all models considered so far can be found in Table 8.

Including an interaction term *ln D · ln X* does slightly increase the average values of Δ*Q_pm_* and *R_A_*^2^. The prediction precisions have improved for the individual observations and are somewhat more homogeneously distributed inside the levels of *X* than in the model (13). Nevertheless, at every orifice level, there is always a particle size fraction for which the predicted mass flow rate has non-ideal precision, i.e., Δ*Q_pm_* is above 10%.

### 3.5. Fully Quadratic Regression Model

This systematic inspection of models aims to propose a simple, accurate model valid for the entire measured range of particle sizes. However, none of the models (12)–(14) for the entire measured particle size range yield satisfactory prediction precisions. Moreover, the question of whether an interaction term *ln D · ln X* has to be included cannot be answered conclusively: The model (14) with interaction is slightly more accurate than model (13), but the interaction term is not statistically significant (and neither is the slope for *ln X* in this model).

We, therefore, at the price of a somewhat more complicated equation, investigate interaction and precision for the complete quadratic model (6). This results in the equation:(15)$$lnQ_{pm}=-0.4\cdot {\left(lnD\right)}^{2}-0.41\cdot {\left(lnX\right)}^{2}+4.77\cdot ln\;D-1.71\cdot lnX+0.27\cdot lnX\cdot \;lnD-7.22$$

Predictions and other model properties are represented in Table 9; the model is illustrated graphically in Figure 6, where with a standard angle of view Figure 6a, the curvature of regression is clearly visible and even more noticeable from the different angles in Figure 6b. The *p*-values for the individual coefficients are given in Table 10.

Not only are all coefficients statistically significant, the adjusted coefficient of determination *R_A_*^2^ is the same quality as for the initial models (8) to (11). Moreover, the average prediction precision is 3.13%, and all precisions, at all levels, are now below 10%. This fully quadratic model gives a very good fit to the measured data. In addition, the estimate of 0.27 obtained for the interaction term is the same as in the previous model (14); however, it is statistically significant in this model.

## 4. Conclusions

In this note, we explored regression models based on the Jones–Pilpel equation with logarithmical transformation to predict the mass flow rate *Q_m_* of a model, free-flowable excipient sorbitol. Simple linear regression on orifice diameter allows predicting *Q_m_* with high precision, as expressed by the average relative deviation of 1.27–5.27% between the experimentally measured and the predicted flow rate, wherever a narrow particle size distribution is tested. If the simple model is used for the broader, entire particle size distribution, the precision of prediction decreases to approximately 14%. Considering the influence of particle size simultaneously through multilinear regression, the prediction of precision increases to 7.5%; this can be further improved to 7% when adding a term for the interaction between orifice and particle diameter. A fully quadratic model achieves the high precision of *Q_m_* prediction of 3.1%.

The complete quadratic model (15) for the logarithmical transformation of *Q_m_* is surprisingly accurate. One may ask whether it corresponds to a model for *Q_m_* of a form familiar in the literature, but to the best of our knowledge, this does not seem to be the case. From (15), we obtain
$$Q_{pm}=e^{-0.4\cdot {\left(ln\;D\right)}^{2}-0.41\cdot {\left(ln\;X\right)}^{2}+4.77\cdot ln\;D-1.71\cdot lnX+0.27\cdot ln\;X\cdot lnD-7.22},$$
which can be written as
$$Q_{pm}=e^{-\left(0.4\cdot {\left(lnD\right)}^{2}-2\cdot 0.135\cdot ln\;X\cdot lnD+0.41\cdot {\left(lnX\right)}^{2}\right)\;}\cdot D^{4.77}\cdot X^{-1.71}\cdot e^{-7.22},$$
where the first factor is reminiscent of the density function for standardized multivariate Gaussian distribution with variables *ln X* and *ln D* (with covariance matrix $\left[\begin{array}{cc} 0.41 & 0.135\\ 0.135 & 0.4 \end{array}\right]$).

Our main conclusions are that (1) for models with satisfactory prediction precision in a broad range of sorbitol particle sizes (0.1 to 0.346 mm), both orifice and particle diameter have to be included as factors in the regression analysis; (2) for highly predictive performance, it is necessary to consider a fully quadratic model; (3) we have found a statistically significant interaction between orifice and particle diameter.

There is certainly more research needed using additional materials. However, the present study based on the model excipient sorbitol for direct compression demonstrated the utility of analysing and modelling the flow behaviour. Sorbitol is a free-flowable substance with a relatively wide particle size distribution. As such materials are quite common in pharmacy, we expect our study’s results will be applicable to materials with similar granulometric characteristics.

## Figures and Tables

**Figure 1 pharmaceutics-14-01653-f001:**
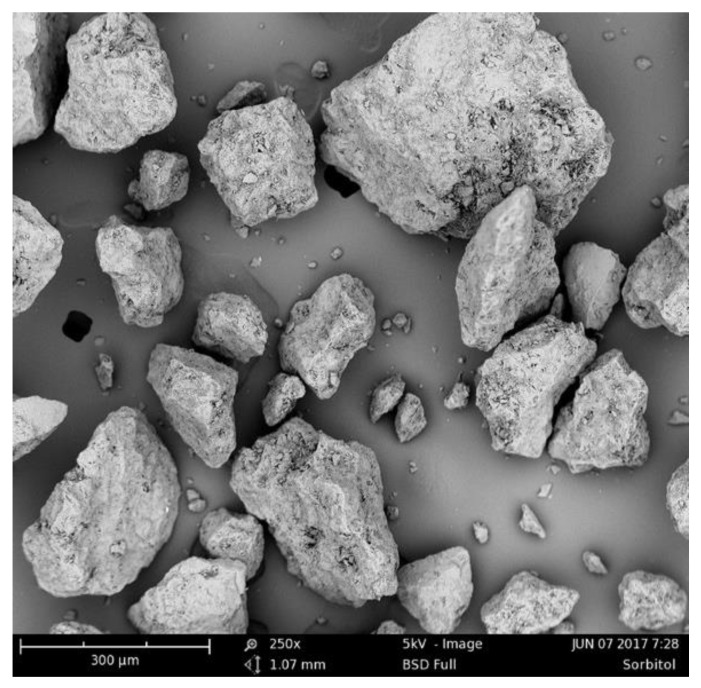
Scanning electron microscope (SEM) image of the raw sorbitol powder before sieving.

**Figure 2 pharmaceutics-14-01653-f002:**
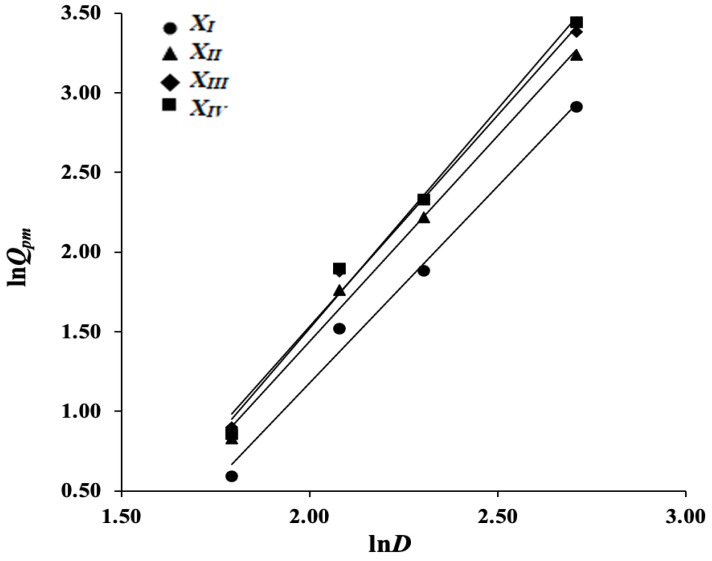
Linear regression lines for *ln Q_pm_* in dependence of *ln D* for each size fraction.

**Figure 3 pharmaceutics-14-01653-f003:**
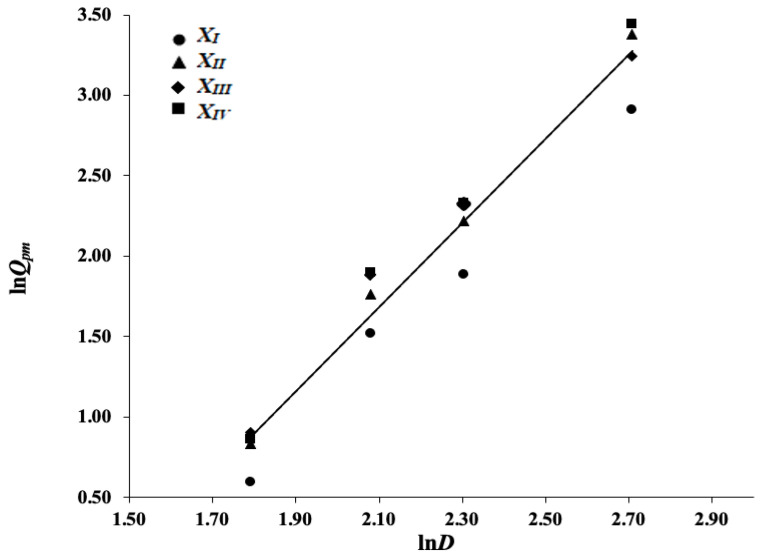
The linear regression line for *ln Q_pm_* in dependence of *ln D* using all particle size fractions pooled.

**Figure 4 pharmaceutics-14-01653-f004:**
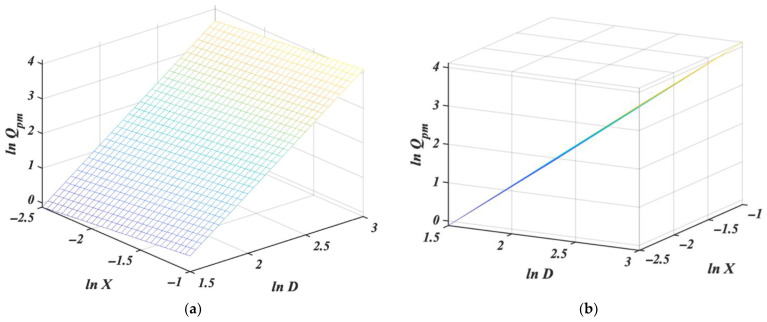
Multiple regression plot for *ln Q_pm_* in dependence of *ln D* and *ln X* without interaction from (**a**) front and (**b**) side of view. Details are given in the text.

**Figure 5 pharmaceutics-14-01653-f005:**
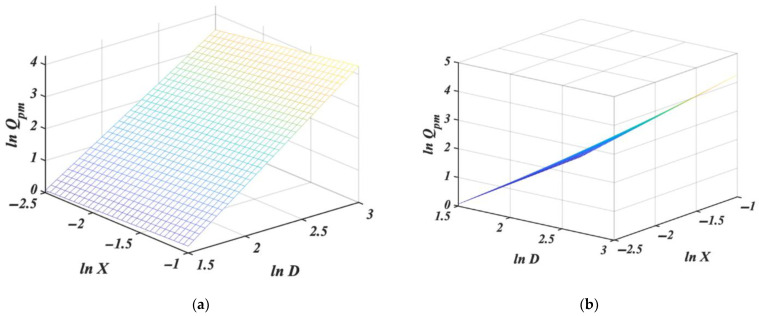
Multiple regression plot for *ln Q_pm_* in dependence of *ln D* and *ln X* with interaction, from (**a**) front and (**b**) side of view. Details are given in the text.

**Figure 6 pharmaceutics-14-01653-f006:**
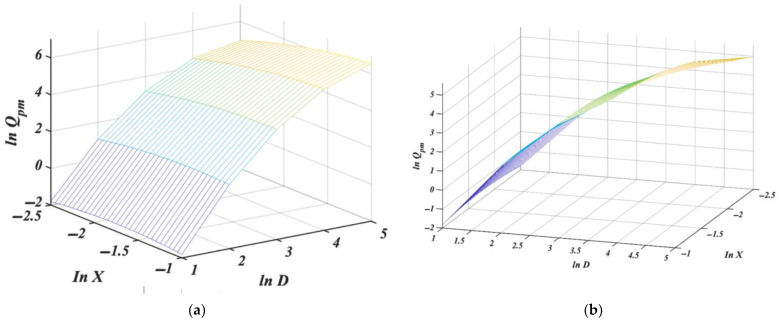
Quadratic regression 3D plot for *ln Q_pm_* in dependence of *ln D* and *ln X* from (**a**) front and (**b**) side of view. Details are given in the text.

**Table 1 pharmaceutics-14-01653-t001:** Notation for particle size levels (SD in brackets).

Level Notation	Particle Size Fraction (mm)	Geometric Mean (mm)	Bulk Density (g/mL)
*X_I_*	0.080–0.125	0.100	0.588 (0.002)
*X_II_*	0.125–0.200	0.158	0.611 (0.003)
*X_III_*	0.200–0.300	0.245	0.619 (0.005)
*X_IV_*	0.300–0.400	0.346	0.639 (0.004)

**Table 2 pharmaceutics-14-01653-t002:** Notation for conical hopper orifice diameter levels.

Level Notation	Corresponding Orifice Size (mm)
*D_I_*	6.0
*D_II_*	8.0
*D_III_*	10.0
*D_IV_*	15.0

**Table 3 pharmaceutics-14-01653-t003:** Experimentally measured mass flow rate (SD in brackets) and calculated volume flow rate.

Particle Size Level	Orifice Level	*Q_m_* (g/s)	*Q_v_* (mL/s)
*X_I_*	*D_I_*	1.96 (0.01)	3.34
	*D_II_*	3.90 (0.08)	6.64
	*D_III_*	6.69 (0.08)	11.39
	*D_IV_*	16.65 (0.17)	28.34
*X_II_*	*D_I_*	2.41 (0.01)	3.95
	*D_II_*	5.37 (0.02)	8.79
	*D_III_*	9.28 (0.03)	15.20
	*D_IV_*	23.37 (0.16)	38.27
*X_III_*	*D_I_*	2.46 (0.02)	3.98
	*D_II_*	5.74 (0.09)	9.28
	*D_III_*	10.33 (0.14)	16.70
	*D_IV_*	27.16 (0.41)	43.90
*X_IV_*	*D_I_*	2.37 (0.01)	3.71
	*D_II_*	5.66 (0.04)	8.85
	*D_III_*	10.39 (0.10)	16.25
	*D_IV_*	27.52 (0.32)	43.04

**Table 4 pharmaceutics-14-01653-t004:** Predictions and quality of models (8)–(11).

Particle Size Level	Orifice Level	*Q_pm_* (g/s)	Δ*Q_pm_* (%)	Average Δ*Q_pm_* (%)	*R* ^2^
*X_I_*	*D_I_*	1.99	1.21	1.27	0.9996
	*D_II_*	3.89	0.31		
	*D_III_*	6.55	2.22		
	*D_IV_*	16.87	1.36		
*X_II_*	*D_I_*	2.53	4.90	4.21	0.9974
	*D_II_*	5.14	4.27		
	*D_III_*	8.91	3.98		
	*D_IV_*	24.24	3.70		
*X_III_*	*D_I_*	2.59	5.29	4.64	0.9972
	*D_II_*	5.49	4.23		
	*D_III_*	9.83	4.83		
	*D_IV_*	28.30	4.20		
*X_IV_*	*D_I_*	2.51	5.96	5.27	0.9965
	*D_II_*	5.40	4.51		
	*D_III_*	9.79	5.75		
	*D_IV_*	28.86	4.86		

**Table 5 pharmaceutics-14-01653-t005:** Predictions and quality of model (12).

Orifice Level	Particle Size Level	*Q_pm_* (g/s)	Δ*Q_pm_* (%)	Average Δ*Q_pm_* (%)	*R* ^2^
*D_I_*	*X_I_*	2.40	21.85	14.26	0.9621
	*X_II_*		0.75		
	*X_III_*		2.96		
	*X_IV_*		0.94		
*D_II_*	*X_I_*	4.93	26.53		
	*X_II_*		8.07		
	*X_III_*		13.97		
	*X_IV_*		12.79		
*D_III_*	*X_I_*	8.66	29.31		
	*X_II_*		6.73		
	*X_III_*		16.18		
	*X_IV_*		16.69		
*D_IV_*	*X_I_*	24.04	44.41		
	*X_II_*		2.86		
	*X_III_*		11.48		
	*X_IV_*		12.65		

**Table 6 pharmaceutics-14-01653-t006:** Predictions and quality of model (13).

Orifice Level	Particle Size Level	*Q_pm_* (g/s)	Δ*Q_pm_* (%)	Average Δ*Q_pm_* (%)	*R* ^2^	*R_A_* ^2^
*D_I_*	*X_I_*	1.97	0.31	7.49	0.9885	0.9868
	*X_II_*	2.26	6.28			
	*X_III_*	2.57	4.51			
	*X_IV_*	2.86	20.56			
*D_II_*	*X_I_*	4.06	4.17			
	*X_II_*	4.66	13.20			
	*X_III_*	5.31	7.34			
	*X_IV_*	5.89	4.16			
*D_III_*	*X_I_*	7.13	6.46			
	*X_II_*	8.17	11.92			
	*X_III_*	9.32	9.73			
	*X_IV_*	10.34	0.49			
*D_IV_*	*X_I_*	19.79	18.89			
	*X_II_*	22.70	2.87			
	*X_III_*	25.89	4.67			
	*X_IV_*	28.71	4.33			

**Table 7 pharmaceutics-14-01653-t007:** Predictions and quality of the model (14).

Orifice Level	Particle Size Level	*Q_pm_* (g/s)	Δ*Q_pm_* (%)	Average Δ*Q_pm_* (%)	*R* ^2^	*R_A_* ^2^
*D_I_*	*X_I_*	2.12	8.25	6.99	0.9910	0.9887
	*X_II_*	2.31	4.16			
	*X_III_*	2.50	1.52			
	*X_IV_*	2.66	12.47			
*D_II_*	*X_I_*	4.17	6.81			
	*X_II_*	4.69	12.55			
	*X_III_*	5.26	8.22			
	*X_IV_*	5.76	1.81			
*D_III_*	*X_I_*	7.02	4.92			
	*X_II_*	8.14	12.30			
	*X_III_*	9.38	9.23			
	*X_IV_*	10.48	0.84			
*D_IV_*	*X_I_*	18.15	9.03			
	*X_II_*	22.13	5.32			
	*X_III_*	26.76	1.47			
	*X_IV_*	31.07	12.91			

**Table 8 pharmaceutics-14-01653-t008:** Coefficient values with statistical significance levels for the models (8)–(14).

	*Model (8)*	*Model (9)*	*Model (10)*	*Model (11)*	*Model (12)*	*Model (13) ^(1)^*	*Model (14) ^(1)^*
** *k* ** ** _0_ **	−3.50 ****	−3.49 ****	−3.49 ***	−3,86 ***	−3.64 ****	−3.15 ****	−4.15 ****
** *k* ** ** _1_ **	2.34 ****	2.47 ****	2.47 ****	2.67 ***	2.52 ****	2.52 ****	2.97 ****
** *k* ** ** _2_ **	-	-	-	-	-	0.30 ****	−0.31
** *k* ** ** _12_ **	-	-	-	-	-	-	0.27

*** = *p* ≤ 0.005; **** = *p* ≤ 0.001, ^(1)^ for the last two models (13) and (14) values of *R_A_*^2^ were computed.

**Table 9 pharmaceutics-14-01653-t009:** Predictions and quality of the model (15).

Orifice Level	Particle Size Level	*Q_pm_* (g/s)	Δ*Q_pm_* (%)	Average Δ*Q_pm_* (%)	*R* ^2^	*R_A_* ^2^
*D_I_*	*X_I_*	2.00	1.95	3.13	0.9995	0.9993
	*X_II_*	2.48	3.00			
	*X_III_*	2.60	5.76			
	*X_IV_*	2.42	1.94			
*D_II_*	*X_I_*	4.22	8.24			
	*X_II_*	5.43	1.19			
	*X_III_*	5.89	2.65			
	*X_IV_*	5.62	0.69			
*D_III_*	*X_I_*	7.20	7.69			
	*X_II_*	9.53	2.73			
	*X_III_*	10.61	2.75			
	*X_IV_*	10.34	0.50			
*D_IV_*	*X_I_*	17.19	3.21			
	*X_II_*	23.91	2.30			
	*X_III_*	27.93	2.83			
	*X_IV_*	28.25	2.64			

**Table 10 pharmaceutics-14-01653-t010:** *p*-values for each term of the full quadratic regression model (15).

*k* _0_	*k* _1_	*k* _2_	*k* _12_	*k* _11_	*k* _22_
1.44 × 10^−9^	1.04 × 10^−8^	3.59 × 10^−7^	2.67 × 10^−5^	6.15 × 10^−5^	3.34 × 10^−7^

## Data Availability

The data presented in this study are available on request from the corresponding author.

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
