# Peer review of "Flow Equations for Free-Flowable Particle Fractions of Sorbitol for Direct Compression: An Exploratory Multiple Regression Analysis of Particle and Orifice Size Influence"

_pharmaceutics, 2022, doi:10.3390/pharmaceutics14081653_

Round 1

Reviewer 1 Report

The only question, which should be addressed by Authors prior publication is the practical applicability of the proposed model. The discussion of influence of patricles geometry on the flowability of the powders should also be taken under consideration.

Reviewer 2 Report

This is a very interesting paper, where the authors developed simple mathematical models for prediction of powder flow rate based on particle size and orifice diameter. It is very important that developed final model is applicable for wide range of particle size.

In order to further support the validity of this model, it will be useful if authors can test model on different powders, since some powder specific factors can also affect flowability.

Minor suggestions:

Please add time duration of sieving process.

Standard deviation should be added to all values that were determined by replicate measurements.

There is an extra space in line 133.
